# Identification of Urine Metabolic Markers of Stroke Risk Using Untargeted Nuclear Magnetic Resonance Analysis

**DOI:** 10.3390/ijms25137436

**Published:** 2024-07-06

**Authors:** Nádia Oliveira, Adriana Sousa, Ana Paula Amaral, Ricardo Conde, Ignacio Verde

**Affiliations:** 1Health Sciences Research Centre (CICS-UBI), University of Beira Interior (UBI), Av. Infante D. Henrique, 6200-506 Covilhã, Portugal; nadiaoliveira@fcsaude.ubi.pt (N.O.); amssousa@fcsaude.ubi.pt (A.S.); anapaula@fcsaude.ubi.pt (A.P.A.); 2Precision Medicine and Metabolism Laboratory, CIC bioGUNE, 48160 Derio, Bizkaia, Spain; ralves@cicbiogune.es

**Keywords:** stroke risk, biomarkers, NMR metabolomic analysis, urine, aging

## Abstract

Stroke remains the second leading cause of mortality worldwide, and the third leading cause of death and morbidity combined, affecting more than 12 million people every year. Stroke pathophysiology results from complex interactions of several risk factors related to age, family history, gender, lifestyle, and the presence of cardiovascular and metabolic diseases. Despite all the evidence, it is not possible to fully prevent stroke onset. In recent years, there has been an exploration of innovative methodologies for metabolite analysis aimed at identifying novel stroke biomarkers. Utilizing Nuclear Magnetic Resonance (NMR) spectroscopy, we investigated small molecule variations in urine across different stages of stroke risk. The Framingham Stroke Risk Score was used in people over 63 years of age living in long-term care facilities (LTCFs) to calculate the probability of suffering a stroke: low stroke risk (LSR, control), moderate stroke risk (MSR), and high stroke risk (HSR). Univariate statistical analysis showed that urinary 4-hydroxyphenylacetate levels increased while glycolate levels decreased across the different stroke risk groups, from the LSR to the HSR groups. Trimethylamine *N*-oxide (TMAO) had average concentration values that were significantly higher in elderly people in the HSR group, while trigonelline levels were significantly lower in the MSR group. These metabolic markers can be used for early detection and to differentiate stages of stroke risk more efficiently.

## 1. Introduction

The definition of stroke orbits around its clinical manifestation; it is a disease characterized by an abrupt onset of neurological impairment resulting from either infarction or haemorrhage in a specific region of the central nervous system [1]. Although the mortality rate attributable to stroke has declined in the past decade, there has been a 26% increase in the absolute number of stroke-related fatalities [1]. According to the latest data, there are over 12.2 million new strokes each year, making stroke the second leading cause of mortality worldwide and the third leading cause when considering both mortality and disability [1,2]. Outstandingly, 66% of deaths occur in individuals aged 70 and above. Furthermore, ischemic stroke, resulting from a blockage in the bloodstream leading to brain tissue damage and subsequent cell death, constitutes approximately 80% of strokes, while haemorrhagic stroke, which occurs when a vessel ruptures and bleeds into the brain and has a higher probability of death, comprises the remaining 20% [2,3]. Investments by health systems in measures to prevent stroke have been increasing annually given that the existing way of preventing it is to detect and control identified stroke risk factors [1,2]. Elevated blood systolic pressure, body mass index, and fasting plasma glucose together with environmental particle pollution and smoking are considered the five leading risk factors for stroke [2]. Additionally, age progression, a nonmodifiable risk factor, also plays a fundamental role in the development of stroke [4]. Other less relevant factors for stroke are family history, dyslipidaemia, sleep-disordered breathing, chronic inflammation, chronic kidney disease, unhealthy diet, alcohol abuse, physical inactivity, depression, and stress [1,5]. Nonetheless, the known risk factors are insufficient for accurately predicting stroke occurrence, maybe due to the complex aetiology of the disease. The emergence of novel biomarkers, potentially serving as new risk factors, can allow a better comprehension of stroke pathophysiology, an improved prognosis, and the rise of new strategies for treatment [6]. Urine biomarkers play a crucial role in the early diagnosis, prognosis, and monitoring of various diseases [7]. Some authors started looking for stroke urine biomarkers in common clinical urine parameters like creatinine, urea, sodium, potassium, estimated glomerular filtration rate (eGFR), high-sensitive cardiac troponin T, and CK proteins [8,9]. As other authors who utilized Nuclear Magnetic Resonance (NMR) signalled, this is a powerful tool in the identification and characterization of urine biomarkers for stroke [10,11]. NMR analysis of urine biomarkers offers advantages such as non-invasiveness, ease of sample collection, and cost-effectiveness [12].

Other authors, using NMR spectroscopy to investigate metabolic alterations during the acute phase of ischemic stroke, documented significant changes in the metabolism and biosynthesis of various metabolites, including phenylalanine, tyrosine, purine, glycerophospholipid, tryptophan, citrate, dimethylamine, creatine, glycine, and hippurate. These metabolites play crucial roles in numerous physiological processes, such as gene expression, neurotransmitter synthesis and metabolism, and response to oxidative stress and neuroinflammation [10,11]. Other authors, using liquid chromatography coupled with mass spectrometry, also found significant associations between ischemic stroke and compounds linked to cell membrane oxidation, formation of oxygen free radicals, neurotransmitter metabolism, and neurotoxicity [13,14].

In this study, we analysed urine samples from individuals with different degrees of stroke risk, using a representative populational sample of users from long-term care facilities. For the analysis, the spectra obtained via 1H-NMR were used to quantify various metabolites present in the urine samples. Additionally, we considered the treatments and concomitant diseases of the participants to study possible influences on the variations in the metabolites analysed. Through univariate statistical analysis, we identified potential metabolic biomarkers capable of distinguishing between different levels of stroke risk in the studied population.

## 2. Results

### 2.1. Personal, Sociodemographic, and Clinical Data

As study groups might display significant differences in personal, sociodemographic, and clinical characteristics, we characterize these parameters by groups because they may have an influence on metabolite levels. Cross-analysis (Table 1, Table 2 and Table 3) was used to summarize various parameters that were subsequently considered.

Concerning gender, the proportion of women exceeds that of men in the MSR and HSR groups (*p* < 0.001). Moreover, compared to the LSR group, the average ages of individuals in the MSR and HSR groups are significantly higher than that of individuals in the LSR group (*p* < 0.001). This is associated with the demographic composition of LTCF residents involved in the study, who are predominantly women aged over 84 years. Furthermore, female individuals have a longer average life expectancy, which certainly contributes to these findings. Clinical measures, such as blood pressure, heart rate, serum cholesterol, and triglyceride levels did not significantly differ between the study groups. Conversely, the average body mass index (BMI) value was substantially higher in the HSR group than in the other groups (*p* = 0.021) (Table 1).

Table 2 shows the frequencies and proportions of the main morbidities by groups. The prevalence of hypertension (*p* = 0.039), acute myocardial infarction (*p* = 0.005), atrial fibrillation (*p* < 0.001), and other arrhythmias (*p* = 0.044) increased significantly from the LSR group to the HSR group (Table 2). Likewise, the total number of cardiovascular diseases also increased from the low-risk group (LSR) to the high-risk group (*p* < 0.001). These results were expected once the FSRS, used to generate the study groups, contemplated in the algorithm the existence of several cardiovascular pathologies (Table 2). No statistically significant differences were found between stroke risk groups for other cardiovascular diseases such as heart failure, angina pectoris, atherosclerosis, valvopathies, or peripheral vascular disease (Table 2). Similarly, the metabolic, respiratory, and central nervous system comorbidities analysed did not show statistically significant differences between the different groups (* *p* < 0.05).

**Table 1 ijms-25-07436-t001:** Characterization of personal, sociodemographic, and clinical information in the low stroke risk (LSR, control), moderate stroke risk (MSR), and high stroke risk (HSR) study groups. Data are expressed in “mean ± s.e.m”, except for gender, which is expressed in “percentage (n)”. Statistical significance was assessed using the Kruskal–Wallis test (KWT), One-way ANOVA test (OWAT), and Fisher’s exact test (FET), depending on the homogeneity of variance and the existence of nominal variables. (* *p* < 0.05).

Conditions	LSR	MSR	HSR	*p*-Value	Statistical Test
Number (n)	73	87	17		
Age (years)	79.5 ± 0.99	88.1 ± 0.60	90.1 ± 1.33	<0.001 *	KWT
Female	49.3 (36)	81.6 (71)	64.7 (11)	<0.001 *	FET
Systolic blood pressure (mmHg)	117.8 ± 2.14	128.7 ± 2.39	132.4 ± 5.44	0.001 *	OWAT
Diastolic blood pressure (mmHg)	68.1 ± 1.45	69.4 ± 1.17	72.4 ± 2.43	0.349	OWAT
Heart rate (bpm)	73.4 ± 1.29	72.0 ± 1.11	71.8 ± 3.01	0.678	OWAT
Body mass index (kg/m^2^)	25.9 ± 0.63	27.8 ± 0.67	29.6 ± 1.20	0.021 *	OWAT
Serum total cholesterol (mg/dL)	164.4 ± 5.80	167.4 ± 4.76	152.5 ± 10.8	0.503	OWAT
Serum HDL cholesterol (mg/dL)	53.7 ± 1.35	56.0 ± 1.39	55.4 ± 3.66	0.518	OWAT
Serum LDL cholesterol (mg/dL)	89.2 ± 5.09	90.2 ± 4.24	75.8 ± 9.36	0.432	OWAT
Serum triglyceride (mg/dL)	107.3 ± 5.03	106.1 ± 5.23	106.5 ± 13.22	0.988	OWAT

**Table 2 ijms-25-07436-t002:** Characterization of the main morbidities in stroke risk groups, namely, low stroke risk (LSR, control, n = 73), moderate stroke risk (MSR, n = 87), and high stroke risk (HSR, n = 17). COPD, Chronic obstructive pulmonary disease. Data are expressed as “percent (n)”. Statistical significance was analysed using Fisher’s exact test (FET) due to the existence of nominal variables. Kruskal–Wallis test (KWT) was used for comparing the number of cardiovascular disease means between groups and data are expressed as “Mean ± s.e.m” (* *p* < 0.05).

Diseases	LSR	MSR	HSR	*p*-Value	Statistical Test
CARDIOVASCULAR DISEASES					
Hypertension	49 (67.1)	74 (85.1)	14 (82.4)	0.039 *	FET
Atrial fibrillation	2 (2.7)	12 (13.8)	8 (47.1)	<0.001 *	FET
Arrhythmia	12 (16.4)	19 (21.8)	8 (52.9)	0.044 *	FET
Heart Failure	11 (15.1)	25 (28.7)	8 (47.1)	0.052	FET
Angina pectoris	2 (2.7)	9 (10.3)	1 (5.9)	0.254	FET
Atherosclerosis	3 (4.1)	4 (4.6)	1 (5.9)	1.000	FET
Valvopathies	0 (0)	4 (4.6)	2 (11.8)	0.112	FET
Peripheral vascular disease	7 (10.3)	7 (8.4)	1 (5.9)	0.924	FET
Acute myocardial infarction	1 (1.4)	1 (1.1)	4 (23.5)	0.005 *	FET
Number of cardiovascular diseases	1.01 ± 0.10	1.40 ± 0.10	2.00 ± 0.26	<0.001 *	KWT
METABOLIC DISEASES					
Diabetes	20 (27.4)	23 (26.4)	7 (41.2)	0.792	FET
Dyslipidaemia	29 (39.7)	39 (44.8)	8 (47.1)	0.768	FET
RESPIRATORY DISEASES					
Chronic obstructive pulmonary disease (COPD)	4 (5.5)	6 (6.9)	1 (5.9)	0.903	FET
Asthma	1 (1.4)	4 (4.6)	1 (5.9)	0.357	FET
CENTRAL NERVOUS SYSTEM DISEASES					
Depression	12 (16.4)	15 (17.2)	1 (5.9)	0.624	FET

Table 3 summarizes the pharmacological classes most used by the individuals participating in this study. Concerning the treatments for cardiovascular diseases, no significant differences were found among stroke risk groups in the proportion of use of several drugs like antiarrhythmics, antianginals, angiotensin-converting enzyme inhibitors (ACEi), angiotensin receptor antagonists, alpha and beta blockers, calcium channel blockers, potassium-sparing and thiazide diuretics, anticoagulants, and venotropics (Table 3). However, we did find a significant correlation between the increase in the risk of stroke and the increase in the percentage of loop diuretic medications (*p* < 0.001) (Table 3).

**Table 3 ijms-25-07436-t003:** Characterization of the main drugs used to treat comorbidities in the elderly in the study groups, namely low stroke risk (LSR, control, n = 85), moderate stroke risk (MSR, n = 94), and high stroke risk (HSR, n = 18). Data are expressed as “percent (n)”. Statistical significance was analysed using Fisher’s exact test (FET) due to the existence of nominal variables (* *p* < 0.05).

Treatments	LSR	MSR	HSR	*p*-Value	Statistical Test
Treatments for cardiovascular diseases					
Antiarrhythmics	2 (2.7)	5 (5.7)	0 (0)	0.299	FET
Antianginal	7 (9.6)	11 (12.6)	1 (5.9)	0.355	FET
ACE inhibitors	9 (12.3)	18 (20.7)	5 (29.4)	0.070	FET
Angiotensin receptor antagonists	25 (34.2)	37 (42.5)	8 (47.1)	0.184	FET
Alpha and Beta blockers	15 (20.5)	14 (16.1)	5 (29.4)	0.175	FET
Calcium channel blockers	18 (24.7)	12 (13.8)	4 (23.5)	0.091	FET
Potassium-sparing diuretics	2 (2.7)	7 (8.0)	1 (5.9)	0.138	FET
Loop diuretics	19 (26.0)	37 (42.5)	13 (76.5)	<0.001 *	FET
Thiazide diuretics	16 (21.9)	20 (23.0)	2 (11.8)	0.344	FET
Venotropics	6 (8.2)	12 (13.8)	2 (11.8)	0.235	FET
Anticoagulants	31 (42.5)	46 (52.9)	11 (64.7)	0.067	FET
Treatments for metabolic diseases					
Sulfonylureas	4 (5.5)	3 (3.4)	2 (11.8)	0.121	FET
Biguanides	10 (13.7)	16 (18.4)	2 (11.8)	0.340	FET
DPP-4 inhibitors	12 (16.4)	11 (12.6)	4 (23.5)	0.197	FET
Insulin	5 (6.8)	3 (3.4)	1 (5.9)	0.211	FET
Statins	28 (38.4)	33 (37.9)	7 (41.2)	0.438	FET
Treatments for respiratory diseases					
Bronchodilators	8 (11.0)	13 (14.9)	2 (11.8)	0.362	FET
Treatments for CNS diseases					
Acetylcholinesterase inhibitors	6 (8.2)	11 (12.6)	1 (5.9)	0.286	FET
Monoamine oxidase inhibitors	1 (1.4)	0 (0)	0 (0)	0.216	FET
N-methyl-D-aspartate antagonist	5 (6.8)	11 (12.6)	1 (5.9)	0.220	FET
Antiepileptics	8 (11.0)	7 (8.0)	1 (5.9)	0.381	FET
Antipsychotics	25 (34.2)	24 (27.6)	4 (23.5)	0.320	FET
Antidepressants	28 (38.4)	32 (36.8)	7 (41.2)	0.923	FET
Treatments for gastric disorders					
Proton pump inhibitors	37 (50.7)	53 (60.9)	13 (76.5)	0.036 *	FET
Laxatives	10 (13.7)	14 (16.1)	3 (17.6)	0.384	FET
Treatments for inflammatory diseases and pain					
Opiates derivatives	12 (16.4)	12 (13.8)	2 (11.8)	0.424	FET
Medicines used to treat gout	7 (9.6)	12 (13.8)	6 (35.3)	0.014 *	FET
Nonsteroidal anti-inflammatory drugs	13 (17.8)	25 (28.7)	7 (41.2)	0.034 *	FET

Regarding treatments for metabolic, respiratory, and central nervous system diseases, no significant differences or correlations were found between the drugs used and the risk of suffering a stroke in the groups under study.

The treatments for gastric disorders, as expected, showed a significant correlation between the increase in stroke risk and the increase in the proportion of proton pump inhibitors (PPi) used (*p* = 0.036) (Table 3). The percentage of individuals taking PPi increases from the LSR group to the HSR group, since these drugs are used to minimize gastric damage caused by taking other drugs, such as nonsteroidal anti-inflammatory drugs (NSAIDs). On the other hand, according to Table 2 and Table 3, individuals in the HSR group have a higher prevalence of various diseases and take several medications, with PPi use being clinically recommended to minimize possible gastric problems associated with polypharmacy.

There is also a significant correlation between the increased risk of stroke and the increase in the use of drugs indicated to treat pain and inflammation, such as medicines used to treat gout (*p* = 0.014) and NSAIDs (*p* = 0.034) (Table 3).

### 2.2. NMR Spectra Peak Integration and Identification

Using NOESY, J-Res, HMBC, and HSQC pulse sequences, 44 metabolites/signals were identified (Figure 1). A customized R code integrated with the AlpsNMR R package version 4.0.4 plus the use of TopSpin^®^ software 3.1 (Bruker, Rheinstetten, Germany, and HMDB/HSQC spectra consultation allowed the identification and integration of the isolated signals of different metabolites from the NOESY spectra (Figure 1).

Spectral data for the signals’ chemical shifts and multiplicity are present in Appendix A.

### 2.3. Univariate Metabolomic Analysis of Metabolite Levels

The absolute concentrations of urine metabolites (mmol/L, mg/dL) detected by NMR spectra analysis of individuals with different levels of stroke risks were examined to identify potential biomarkers. Univariate statistical analysis was used to identify these biomarkers, which are presented in Appendix A.

We observed significant differences between the stroke risk groups in the concentrations of five types of metabolites, namely, trimethylamine *N*-oxide (TMAO), total sugar, glycolate, 4-hydroxyphenylacetate (4-HPA), and trigonelline (Figure 2).

TMAO levels showed significant changes with increasing risk of stroke (*p* = 0.042). The statistical analysis showed that TMAO levels in the HSR group are significantly superior to levels in the MSR group (Figure 2A). TMAO levels are not significantly different (*p* = 0.090) between the LSR and MSR groups, and the concentrations are similar in the LSR and HSR groups (*p* = 0.288). We also found a significant decrease in sugar (*p* = 0.050) levels in the LSR and MSR groups (Figure 2B). Sugar concentrations did not vary between the LSR and HSR groups. For glycolate, a significantly lower mean concentration was found in the HSR group compared with the LSR and MSR groups (*p* = 0.040 and 0.014, respectively). For LSR and MSR pair analysis, no significant differences were found (*p* = 0.534). We found a significant increase in 4-HPA concentrations with increasing stroke risk across the study groups, including between LSR and MSR (*p* = 0.040) as well as MSR and HSR (*p* = 0.014). However, no meaningful differences were found for the increase in concentration between LSR and HSR groups (*p* = 0.248). We also found a decrease in trigonelline concentration with increasing stroke risk across the study groups. This decrease was statistically significant between the LSR and MSR groups (*p* = 0.022), but not between the LSR and HSR groups (*p* = 0.130) or between the MSR and HSR groups (*p* = 0. 825).

The discriminatory power of the stroke risk of the five metabolites that showed significant differences between study groups was estimated using analysis of the area under the receiver operating characteristic curve (AUROC; sensitivity/specificity) using absolute concentrations obtained from the univariate analysis. Glycolate (*p* = 0.010) had an AUROC of 0.675. Total sugar (*p* = 0.159), trimethylamine *N*-oxide (*p* = 0.165), and 4-HPA (*p* = 0.070) had AUROCs between 0.600 and 0.650, while the trigonelline (*p* = 0.428) ROC area was below 0.600. Trimethylamine *N*-oxide, total sugar, 4-HPA, and trigonelline had AUROC values under 0.650 and nonsignificant *p*-values, indicating lower discriminatory power for stroke risk (Figure 3). Thus, these data indicate good performance with good AUROC parameters for glycolate (Figure 3).

### 2.4. Comorbidities and Drug Effects on Urine Metabolome

Further statistical analyses were conducted to investigate the possible influence of certain diseases and pharmacological treatments on previously identified variations in metabolite levels across the different stroke risk categories. Our focus in this analysis was on diseases and therapies exhibiting distinct prevalence and utilization rates, respectively, across the study groups (Table 2 and Table 3).

Given the significant increase in the average age and total number of cardiovascular diseases observed among stroke risk groups (Table 1), we performed a correlation analysis between age and the number of cardiovascular diseases, and the urine concentrations of TMAO, total sugar, glycolate, 4-HPA, and trigonelline in all the studied people, as well as among the individuals belonging to each stroke risk group (Table 4). A significant correlation was observed between the levels of 4-HPA and age (r = 0.162; *p* = 0.045), and 4-HPA and the total number of cardiovascular diseases (r = 0.178; *p* = 0.030) (Table 4). Although significant, considering the low values of the correlation coefficients in both cases, age and the total number of cardiovascular diseases do not appear to be confounding variables in the observed differences in urine metabolite levels across stroke risk groups. We also found a significant negative correlation between trigonelline levels and the total number of cardiovascular diseases (r = −0.258; *p* < 0.001). This leads us to believe that the decrease in trigonelline levels may be influenced by the presence of multiple cardiovascular diseases.

Moreover, as depicted in Table 2, a noteworthy positive correlation was evident between individuals diagnosed with arrhythmia and an increasing risk of stroke. We conducted a comparison of the mean values of TMAO, total sugar, glycolate, 4-HPA, and trigonelline between individuals with and without arrhythmia, both across all individuals and within each stroke risk group. Concerning atrial fibrillation, we found that mean trigonelline levels are significantly lower in people with this disease (*p* < 0.001) (Table 5). Examining each of the study groups, trigonelline does not show any significant correlation with atrial fibrillation (Table 5); hence, atrial fibrillation appears to have a positive effect on reducing concentrations of trigonelline.

Some pharmacological treatments showed a positive correlation with increased risk of stroke (Table 3). In this sense, individuals treated with loop diuretics (LoDiu) increased from the LSR to the HSR groups (Table 3), and we found a significant increase in total sugar (*p* = 0.046) and 4-HPA (*p* = 0.025) in people taking them. Additionally, we also found positive and significant, although weak, correlations between the percentage of elderly people in the MSR group and total sugar (r = 0.225; *p* = 0.037) and 4-HPA (r = 0.251; *p* = 0.030). On the other hand, these results did not reflect the significant decrease found in total sugar levels in the MRS group (Figure 2). Therefore, LoDiu treatment does not seem to influence the levels of this metabolite.

The significantly higher levels of 4-HPA in people taking LoDiu and the increased concentrations in the MSR and HSR groups (Figure 2) suggest that the use of this drug appears to positively influence the increase in 4-HPA.

The utilization of proton pump inhibitors (PPi) increases significantly from the LSR to the HSR groups (Table 3). We compared the average values of metabolites between people who take and do not take PPi and observed that the mean levels of total sugar (*p* = 0.044) and glycolate (*p* = 0.003) are significantly higher in individuals who use these drugs (Table 5). The analysis carried out on individuals in each of the study groups showed that the variations in total sugar and glycolate levels do not significantly correlate with the use of PPi (Table 5). Furthermore, the significant increase in glycolate concentrations in people taking PPi is contradictory to the decreasing levels of this metabolite as the risk of stroke increases and so does the decrease in total sugar levels in the elderly in the MSR group as opposed to the increased prevalence of taking this drug. Thus, the use of PPi does not appear to influence the decrease in glycolate and total sugar associated with increased risk of stroke (FSRS).

Likewise, the use of medicines to treat gout (colchicine and allopurinol), drugs used to reduce uric acid production, also increased with the increased risk of stroke (Table 3). We found a significant decrease in the levels of glycolate in individuals medicated with these drugs (*p* = 0.003) (Table 5). Still, while analysing the correlation in each stroke risk group, we did not find a significant correlation between these medicines and glycolate levels (Table 5). Hence, the use of medicines to treat gout appears to positively influence the decrease in glycolate levels associated with the increased risk of stroke (FSRS).

The rate of use of non-steroidal anti-inflammatory drugs (NSAIDs) increases significantly with the increase in the risk of stroke by group (Table 3). We compared the mean metabolite values between individuals using or not using NSAIDs as therapy and observed that average 4-HPA levels are significantly higher in individuals treated with these drugs (*p* = 0.036) (Table 5). Thus, the use of NSAIDs appears to positively influence the increase in 4-HPA associated with increased stroke risk (FSRS). However, no significant correlations were observed between NSAID intake and 4-HPA levels in either study group (Table 5).

Summarizing the effects of drugs as confounding factors, we can state that the existence of weak correlation parameters, the lack of correlation in various stroke risk groups, and the contradictory data indicate that the previously mentioned drugs have weak or no influence on the urine levels of total sugar, glycolate, 4-HPA, and trigonelline, with the exception of medicines used to treat gout and glycolate, and LoDiu/NSAIDs and 4-HPA.

## 3. Discussion

Currently, the identification of urine metabolomic biomarkers specific to stroke remains elusive. Although some studies have been published that point to possible urinary biomarkers, these are far from being concordant or conclusive. Inflammatory mediators, transport and matrix proteins, and small circulating molecules have been reported as possible stroke biomarkers [15,16,17]. These studies focused on finding biomarkers for stroke by comparing healthy individuals with patients in the acute phase of stroke or who were receiving post-stroke hospital care, in populations with different characteristics (age, race, etc.). Moreover, potential confounding factors, such as medication regimens and comorbidities, were frequently disregarded or inadequately assessed. To overcome these limitations, we analysed the urine of elderly people with different degrees of stroke risk in a demographically representative sample.

Employing proton nuclear magnetic resonance (1H-NMR) spectroscopy, we acquired the urine spectra, which enabled the quantification of several metabolites; we also considered the treatments and concomitant diseases of the participants. Through univariate statistical analysis, we identified potential metabolic biomarkers capable of differentiating stroke risk levels. We found that TMAO, total sugar, glycolate, and trigonelline levels are downregulated while 4-HPA is upregulated with increasing stroke risk (FSRS).

We found a significant decrease in total sugar levels (*p* = 0.050) of individuals in the MSR group compared with the control LSR group (LSR, Figure 2B). Total sugar concentration values did not vary between the LSR and HSR groups. The sugar detected in the study comprised many molecules, including monosaccharides (glucose, fructose, galactose) and disaccharides (sucrose, maltose, lactose), and together they are often referred to as “total sugar” or simply “sugars”. Blood sugar levels depend fundamentally on the diet (food sweeteners, fruits, vegetables, and milk), with sweeteners from processed foods currently being among the most significant contributors to sugar intake. Thus, the quantity of sugars in urine is probably a fraction of the sugars in the diet that escapes or arises from metabolic processes and escapes reabsorption in the renal tubules [18]. Urinary sugars have been used as biomarkers of food consumption. For example, according to different authors, obese people consume significantly more sugars than individuals of normal weight [19]. The measurement of sugars using NMR involves the detection of chemically different protons existing in different sugars, which causes complex spectral patterns and signal overlap and complicates the reliable identification and quantification of these molecules. Some authors have reported that this type of compound can be measured using broadband homonuclear decoupling protocols, which may be suitable for concentrated samples as it has low sensitivity and requires long measurement times. On the other hand, there are large inter-individual variations in the concentrations of these sugars in urine, which is fundamentally conditioned by individual intake of water and food in different people [20]. Therefore, in this study, we cannot draw significant conclusions from the differences observed in the average levels of total sugars in stroke risk groups, a subject that deserves a specific study of these compounds but should probably focus on the influence of diet on stroke risk.

In the human body, the majority of trimethylamine is obtained mainly from the diet (red meat, eggs, and poultry), with the rest coming from ammonia metabolism. Choline, L-carnitine, betaine, phosphatidylcholine, and other choline-containing compounds can be converted to trimethylamine by gut microbiota in the colon, and after gut absorption, it can be further converted to its main metabolite trimethylamine *N*-oxide (TMAO) in the liver [21]. Approximately half of the formed TMAO is excreted, and the other half is reduced back to TMAO via the action of TMAO reductase [22]. Different evidence supports that increased TMAO plasma levels are associated with several stroke risk factors, including coronary heart disease, myocardial infarction, hypertension, and atherosclerosis, as well as with stroke itself [22,23,24,25,26,27]. The adverse effects of increased TMAO concentrations on cardiovascular function were hypothesised through multiple mechanisms, both in clinical and preclinical studies [28,29]. The main atherogenic mechanism is based on the increased expression of macrophage scavenger receptors and the formation of foam cells in the arterial wall. It has also been proven that TMAO plays a fundamental role in endothelial dysfunction, exacerbation of platelet reactivity, increase in thrombosis, impairment of lipid metabolism, and modelling of the inflammatory response [22,23,24,25,26,27,28,30]. Regarding these hypotheses, in vitro and in vivo studies showed that TMAO directly promotes and amplifies the stimulation of platelet activation, triggering multiple agonists of this process by increasing the release of intracellular calcium and promoting the accumulation of cholesterol in macrophages by increasing cell surface expression of pro-atherogenic receptors [25,26].

Two case–control studies of a Chinese population confirmed that higher serum TMAO levels were associated with an increased risk of first stroke [27,28]. Nie et al. associated higher TMAO serum levels with the increased risk of the first stroke in a hypertensive Chinese population. This study used liquid chromatography-tandem mass spectrometry (LC-MS) to quantify TMAO and included 622 patients with the first stroke and the same number of controls, overall, with an average age of 62 years [27]. Similarly, Rexidamu and collaborators, in a study involving 255 Chinese patients suffering from acute ischemic stroke, alongside 255 controls with a median age of 65, analysed serum TMAO concentrations using ultra-high-performance liquid chromatography-tandem mass spectrometry (UHPLC-MS/MS). The results revealed a significant association between elevated TMAO levels and increased risk of initial ischemic stroke. Additionally, higher TMAO levels were also correlated with worse neurological deficits in these individuals [28]. Furthermore, Rozzen et al., in a large community-based prospective cohort study with a 15-year follow-up that included 11785 participants with no history of stroke and an average age of 66 years observed that elevated plasma TMAO levels (LC-MS) were associated with a higher risk of ischemic stroke [29]. To date, only one study has focused on the detection and quantification of TMAO in urine. Danxia et al. evaluated urinary levels of TMAO in relation to the risk of coronary heart disease (CHD) in a case–control study including 275 patients and 275 matched controls (~age of 62 years) using UHPLC-MS/MS and concluded that higher urinary levels of this metabolite were associated with increased risk of CHD [31]. Furthermore, these authors also concluded that the presence of diabetes, obesity, and dyslipidaemia, associated with high levels of urinary TMAO, contributed to an increased risk of CHD [31]

Our data show a significant increase in TMAO levels in the HSR group compared with the MSR group, and a nonsignificant decrease between the LSR and MSR groups. These data can be seen as complementary to the studies mentioned above, since urinary TMAO levels may be reduced due to its decreased excretion and consequent retention in blood. However, due to the type of population under study, urinary TMAO excretion may naturally decrease due to renal ageing. Taking these variations into account, it is not uncommon for AUROC analysis to suggest poor TMAO performance when discriminating high stroke risk. To the best of our knowledge, no metabolomic study has been published that reports the association between a decrease in TMAO levels in urine and an increased risk of suffering a stroke. Only one case–control study, carried out by Jia Yin and collaborators, using LC-MS on plasma collected from 553 individuals (average age of between 56 and 61 years) reported that decreased levels of plasma TMAO were associated with stroke occurrence, but not with atherosclerosis [32]. Our study also did not find any correlation between TMAO levels and any of the stroke risk factors analysed.

Glycolate is the smallest alpha-hydroxy acid and, in the human body, comes mostly from the diet (sugarcane, pineapple, and sugar beets), although it can also be endogenously produced through glyoxylate reduction, previously originated from Krebs cycle and collagen biosynthesis [33]. Nowadays, one of the main systemic absorption routes for glycolate comes from its widespread use in skincare and haircare products [34,35]. In vivo and in vitro studies showed that glycolate may modulate oxidative stress, inflammation, and glutamate neurotransmission, which are mechanisms implicated in the pathogenesis of stroke [36,37]. Chovsepian and her colleagues showed the protective effect of glycolate in stroke, by reducing intracellular Ca^2+^ concentrations in an in vitro model of cortical neurons. These authors proved that glutamate-mediated excitotoxicity is implicated in stroke and ischemia-reperfusion injury by elevating intracellular Ca^2+^ levels, leading to cellular dysfunction and apoptosis activation [36]. Moreover, the same authors also confirmed these strong neuroprotective properties of glycolate in a mouse stroke model and in a swine model, in which intravenous glycolate treatment was administered after ischemic injury. Overall, both models revealed a strong, positive effect on neuronal survival after treatment, highlighting the important role of glycolate in post-stroke tissue reperfusion [36]. Other studies used glycolate to produce nanoparticles capable of encapsulating experimental drugs tested to treat stroke due to glycolate’s capacity to inhibit the activation of inflammatory pathways, suppress the expression of pro-inflammatory cytokines, and modulate cellular apoptosis [37,38,39]. Based on this, it is possible to conclude that glycolate is responsible for the inhibition of several processes associated with brain damage, and some of them are also considered pathophysiological pathways of stroke. Our data show that glycolate urine levels decrease with increasing risk of stroke, leading us to conclude that the neuroprotective mechanisms evidenced by glycolate may be compromised. These findings are supported by the AUROC analysis, suggesting a good performance of this metabolite to discriminate high stroke risk. However, to our knowledge, this is the first study to point to glycolate as a biomarker of stroke risk. Further studies are needed to clarify the clinical relevance of low levels of urinary glycolate. We also found a significant correlation between lower glycolate levels and taking medication for gout treatment. In fact, some studies associated high glycolate serum levels with renal failure caused by the accumulation of calcium oxalate crystals in the kidneys [40,41]. Allopurinol is the drug used for chronic gout treatment and is responsible for the decrease in the production of calcium oxalate crystals at a systemic level. Based on this, we believe that allopurinol may be responsible for the lower glycolate urinary levels in the few people taking medicine used to treat gout.

Trigonelline plasma levels also decrease with increasing risk of stroke. Trigonelline is not naturally synthesized in the human body. It is obtained through the ingestion of several plants, including fenugreek seeds, coffee beans, garden peas, hemp seeds, and oats [42,43]. It is already proven that it possesses a range of pharmacological attributes, including anti-apoptotic, anti-inflammatory, antioxidant, anti-diabetic, and neuroprotective properties [42,44,45,46]. It has also been documented to have a protective effect in ischemic stroke, but the precise mechanism behind it remains unknown [43]. Qiu et al. used an in vitro model of oxygen–glucose deprivation/reperfusion with primary hippocampal neuronal mouse cells and treated them with trigonelline. The authors concluded that trigonelline can protect the hippocampal neurons from oxidative stress and mediate neuroinflammation via an increase in SOD activity and GSH levels, a decrease in lipid peroxidation and caspase-3 activity, and TNF-α, IL-6, IL-1β, and Bcl-2-associated protein X levels [45]. Other authors have also proven the ability of trigonelline to interact with myeloperoxidase (MPO; a well-known stroke-related inflammation protein), blocking its action and protecting the brain from MPO-mediated inflammation after stroke [44]. Our data show a significant decrease in trigonelline urinary levels from the LSR to the MSR groups and a nonsignificant decrease in these levels from the LSR to HSR groups. Taking this variation in levels into account, it is not uncommon for AUROC analysis to suggest poor trigonelline performance in discriminating high stroke risk. Despite this, it is possible to state that our results agree with the findings of other authors since the decrease in trigonelline appears to be highly detrimental to a series of protective brain mechanisms for which this molecule is responsible. We also observed decreased trigonelline levels in elderly people with atrial fibrillation, which is the major cardiac comorbidity as a risk factor for stroke. Similarly, we also found a significant correlation between low trigonelline levels and the presence of multiple cardiovascular diseases. This suggests that low trigonelline levels may constitute a differentiating biomarker for the increased risk of stroke in people suffering from atrial fibrillation and multiple cardiovascular diseases. Nevertheless, regarding cardiac effects, the cardioprotective effects of trigonelline were only studied by Ilavenil et al. in an in vitro model of rat cardiac cells. These authors demonstrated that trigonelline has significant protective effects on cardiac cells, notably reducing necrosis and apoptosis caused by hydrogen peroxide, due to increased antioxidant activity and regulation of genes involved in apoptosis such as caspase-3, caspase-9, Bcl-2, and Bcl-XL during oxidative stress [47]. It is well known that oxidative stress is a critical factor in the onset and progression of atrial fibrillation. This reinforces our hypothesis that the decrease in trigonelline levels can be seen as biomarkers of stroke, especially in individuals with atrial fibrillation.

4-hydroxyphenylacetate (4-HPA) is a phenolic acid formed by microbial metabolism of aromatic amino acids (phenylalanine and tyrosine) and polyphenolic compounds, mostly flavonoids, by gut microbiota [48,49]. Tyrosine is converted to 4-hydroxyphenylpyruvate (4-HPP) by the enzyme tyrosine aminotransferase. Next, 4-HPP is transformed into 4-HPA by the enzyme 4-hydroxyphenylpyruvate dioxygenase [49]. This process occurs mostly in the liver, after the absorption of the primary compounds in the colon. Some in vivo studies have already shown 4-HPA hepatoprotective anti-inflammatory properties [48,50]. Liu and collaborators showed that 4-HPA was able to reduce the levels of the pro-inflammatory cytokines TNF-α, IL-1β, IL-6, and HIF-1α and prevent lung oedema in a rat model of acute lung injury [48]. Similarly, using a mouse model of acute liver injury induced by acetaminophen, Zhao et al. showed that 4-HPA was responsible for the up-regulation of antioxidant enzymes and activation of hepatoprotective mechanisms [50]. Concerning clinical studies, Nishiumi and colleagues conducted a metabolomic study (GC-MS) on serum samples collected from 20 patients at different stages of pancreatic adenocarcinoma and compared them with 9 samples from healthy individuals. These authors identified 18 altered metabolites, one of which was 4-HPA, whose levels were significantly increased in patients with stage III and IVb disease. We did not find any studies that addressed the effects of 4-HPA on brain and cardiovascular health in humans. Our results show a significant increase in 4-HPA in elderly people from the LSR to the MSR groups and from the MSR to the HSR groups, but not from the LSR to the HSR groups. These results are supported by the poor power values of the AUROC analysis used to discriminate high stroke risk. Our data also showed a significant correlation between increased 4-HPA levels and pharmacological therapies with loop diuretics and NSAIDs. Based on this, and due to the lack of scientific evidence, it is premature to analyse the importance of increased 4-HPA in relation to the increased risk of stroke. To the best of our knowledge, this is the first study to point to 4-HPA as a biomarker of stroke risk and further studies are needed to clarify the role of this metabolite in cardiac and brain diseases.

In conclusion, the urine metabolite markers of stroke risk described can be used for early detection of disease risk, early diagnosis, and exploration of pathological mechanisms. TMAO levels seem to be a promising biomarker for distinguishing HSR from MSR, indicating a potential link to intestinal microbiome metabolism and elevated plasma TMAO levels. Additionally, our data suggest that glycolate and trigonelline levels decrease with increasing stroke risk, with glycolate showing promise as a discriminating biomarker for stroke risk. Trigonelline levels also correlate with atrial fibrillation and multiple cardiovascular diseases, suggesting its potential as a biomarker in these populations. The decrease in the urinary levels of these two metabolites suggests less protection against pro-inflammatory, neuroexcitatory, and oxygen-free radical formation mechanisms. On the contrary, urinary 4-HPA levels increase with stroke risk, but its higher levels also correlate with loop diuretics and NSAID intake, raising questions about its potential as a biomarker in this context. To the best of our knowledge, this is the first study demonstrating lower urinary levels of TMAO, glycolate, and trigonelline and higher levels of 4-HPA as biomarkers of stroke risk. Given AUROC analysis of TMAO, trigonelline, and 4-HPA, further research is needed, especially to fully understand the significance of these findings and their applicability as potential novel biomarkers for stroke risk management.

It is necessary to emphasize that this study has certain limitations. The number of elders included in the study was relatively small. Thus, a similar study should be conducted on a larger cohort to clarify and strengthen these results. Long-term measurements of dynamics should also be performed to determine what happens over time. Moreover, we could not determine whether the variations in urine metabolite levels are associated with changes in blood or cerebral spinal fluid. Whether or not atrial fibrillation, chronic therapy for gout treatment, loop diuretics and NSAIDs can influence trigonelline, glycolate, and urine 4-HPA concentrations, respectively, should also be examined further.

## 4. Materials and Methods

### 4.1. Study Group Constitution

This study focused on EBIcohort individuals aged 64 years and above, residing in Long-Term Care Facilities (LTCFs) within an approximately 1000 square kilometre area spanning three municipalities in Beira Interior, Portugal [51].

All procedures undertaken were subject to prior review and approval by the Ethics Committee of the University of Beira Interior and were conducted in accordance with the principles of the Helsinki Convention (Reference Number CE-UBI-Pj-2017-012). Written informed consent was obtained from either cohort participants or their legal representatives. Demographic, clinical, and therapeutic data for each participant were provided by the clinical staff of collaborating LTCFs or community health services. Body mass index (BMI; kg/m^2^) was calculated based on anthropometric measurements.

The exclusion criteria for EBIcohort participants applied in this study were: (1) cognitive impairment stemming from trauma, infection, or other causes of brain dysfunction; (2) diagnosis of psychiatric disorders impacting brain function; (3) undertaking aggressive pharmacotherapy with antipsychotics, anticonvulsants, antiretroviral therapy, or antiemetics; (4) clinical history with precedent stroke crisis.

Study groups were stratified based on Framingham Stroke Risk Score (FSRS) values. The FSRS for each participant was computed using clinical data within a 5-year timeframe to estimate stroke probability [52,53]. The FSRS, adjusted in 2017, incorporated modifiable and non-modifiable stroke risk factors, including age, sex, systolic blood pressure, antihypertensive medication usage, presence of cardiovascular diseases, diabetes, and tobacco use history [53]. Participants were categorized into three risk groups: low stroke risk (<5%; LSR or control), moderate stroke risk (5–20%; MSR), and high stroke risk (>20%; HSR).

Serum samples were analysed for various biochemical parameters commonly assessed in healthcare settings using kits optimized for this purpose (BioSystems Human, Barcelone, Spain), including cholesterol oxidase/peroxidase, direct detergent HDL cholesterol, and glycerol phosphate oxidase/peroxidase. LDL values were derived using the Friedewald formula.

### 4.2. Sample Preparation

First-morning void urine samples from overnight-fasted individuals were collected in appropriate cups and refrigerated at 4–8 °C. Within a timeframe of 2 h, four urine aliquots from each participant (1300 µL) were centrifuged for 20 min at 3000× *g* and 4 °C. Supernatant (1 mL) was collected from each tube and stored at −20 °C.

To perform the NMR analysis, 750 µL of thawed urine samples were centrifuged for 5 min at 4 °C and 8000× *g*, and 630 µL of the resultant supernatant was mixed with 70 µL of buffer solution (1.5M KH2PO2 in D2O + 0.1% TSP + 0.1% NaN3). The solution pH was adjusted to 7± 0.02 using KOD (4M) or DCl (4M) as needed. A final centrifugation was performed for 5 min at 4 °C and 8000× *g*, and 600 µL of the resultant supernatant was placed in NMR tubes (5 mm).

### 4.3. NMR Data Acquisition and Processing

Urine NMR spectra were acquired at 300 K on an AVANCE III 600 MHz NMR spectrometer equipped with a quadruple resonance cryoprobe with an automated sample changer (Bruker SampleJet; Bruker, Rheinstetten, Germany). For each urine sample, 1D pulse sequence overhauser effect spectrum with pre-saturation (1D NOESY-presat; noesygppr1d, Bruker library, Bruker, Rheinstetten, Germany) was acquired using 32 scans, 98.304 data points, a spectral width of 20.0286 ppm, acquisition time of 2.72 s, relaxation delay of 4 s, 4 µs T1 delay, and mixing time of 100 ms, with water peak suppression.

The NMR spectra were automatically phased, and baseline corrected and referenced to the internal standard TSP compound (0 ppm) using TopSpin 3.1 software (Bruker, Rheinstetten, Germany).

Two-dimensional (2D) 1H-1H J-resolved (J-res) spectra were also obtained to enhance metabolite identification. TopSpin 3.1 software was used to control the spectrometer and for data pre-processing (Bruker Biospin; Rheinstetten, Germany).

Additionally, two-dimensional (2D) NMR spectra such as 1H-13C Heteronuclear Single-Quantum Correlation Spectroscopy (HSQC) and Heteronuclear Multiple Bond Correlation (HMBC) were acquired for the representative pool sample to aid in the identification of metabolites.

### 4.4. Statistical Analysis

NMR data were managed using R statistical software and processed using the AlpsNMR R package 4.0.4. (R Foundation for Statistical Computing, Vienna, Austria). To avoid the introduction of potential confounder factors in statistical analysis, we excluded the water region between 4.4 ppm and 5.2 ppm. To perform univariate analysis, we used 1D NOESY-presat spectra. We detected and normalized spectral peaks by Probabilistic Quotient Normalization (PQN). Spectral peak assignment and metabolite identification were carried out by matching chemical shift and peak multiplicity with information from the literature and the Human Metabolome Database (HMDB) [12].

After assignment, NMR peaks were integrated separately into metabolite signals with good definition. The TSP and the metabolite signals were integrated and normalized according to the corresponding number of protons. The integrals of the metabolites were then compared with those of the standard compound. The concentration of each metabolite (Cx) in the presence of TSP was calculated using the following formula:

Cx = (Ix/Istd) × (Nstd/Nx) × Cstd, where I is the integral area, N is the number of nuclei, and C is the concentration of the metabolite (x) and the internal standard (std), respectively.

After calculating the metabolite concentrations, the outliers were detected and excluded. The resulting data were analysed using the One-way ANOVA test (OWAT; IBM Corp. released 2021, IBM SPSS Statistics for Windows, Version 28.0. Armonk, NY, USA: IBM Corp) to determine the existence of different average concentrations between the study groups. In cases where the homogeneity of variances was not verified using the Levene test, data were analysed using the Kruskal–Wallis non-parametric test (KWT).

To analyse and compare demographic and clinical data such as gender, age, BMI, systolic blood pressure, comorbidities, and medication, we have applied OWAT, KWT, or Fisher’s exact test (FET). To examine differences in the metabolite levels in the presence or absence of other variables or correlations between different variables and detect possible confounders, we used the Student’s T-test or Spearman’s rank correlation (SRC), respectively. Box–violin–scatter plot representatives of the results of metabolite concentrations were generated. Analysis of the area under the receiver operating characteristic curve (AUROC) was performed to validate the possible biomarkers (IBM Corp. released 2021, IBM SPSS Statistics for Windows, version 28.0. Armonk, NY, USA: IBM Corp).

## Figures and Tables

**Figure 1 ijms-25-07436-f001:**
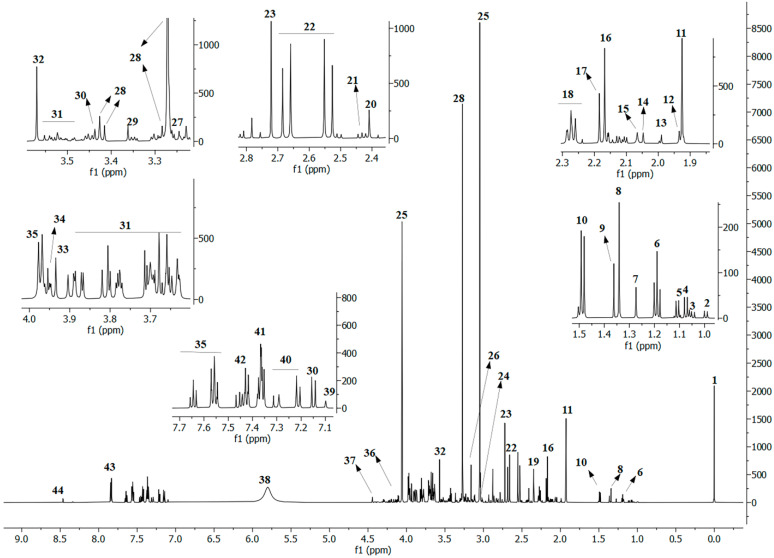
600 MHz 1H-NMR NOESY spectra of urine pooled samples. Numbers indicate the following metabolites: 1—TSP, 2—Isoleucine, 3—Valine, 4—Isobutyrate, 5—Methylsuccinate, 6—Ethanol, 7—3-hydroxyisovalerate (3-HIVA), 8—Threonine, 9—alpha-hydroxyisobutyrate (2-HIBA), 10—Alanine, 11—Lysine, 12—Acetate, 13—2-hydroxyglutarate, 14—N-acetylaspartate, 15—N-acetylneuraminate, 16—Acetone, 17—Acetoacetate, 18—Glutamate, 19—Pyruvate, 20—Succinate, 21—Glutamine, 22—Citrate, 23—Dimethylamine, 24—Creatine, 25—Creatinine, 26—Methylurate, 27—Trimethylamine *N*-oxide (TMAO), 28—Taurine, 29—Methanol, 30—4-hydroxyphenylacetate (4-HPA), 31—Sugar (glucose, sucrose, manitol, galactose, xylose), 32—Glycine, 33—Glycolate, 34—Serine, 35—Hippurate, 36—Lactate, 37—Trigonelline, 38—Urea, 39—Histidine, 40—Indoxyl sulphate, 41—Phenylacetylglutamine (PAG), 42—Phenylalanine, 43—Nudifloramide, 44—Formate.

**Figure 2 ijms-25-07436-f002:**
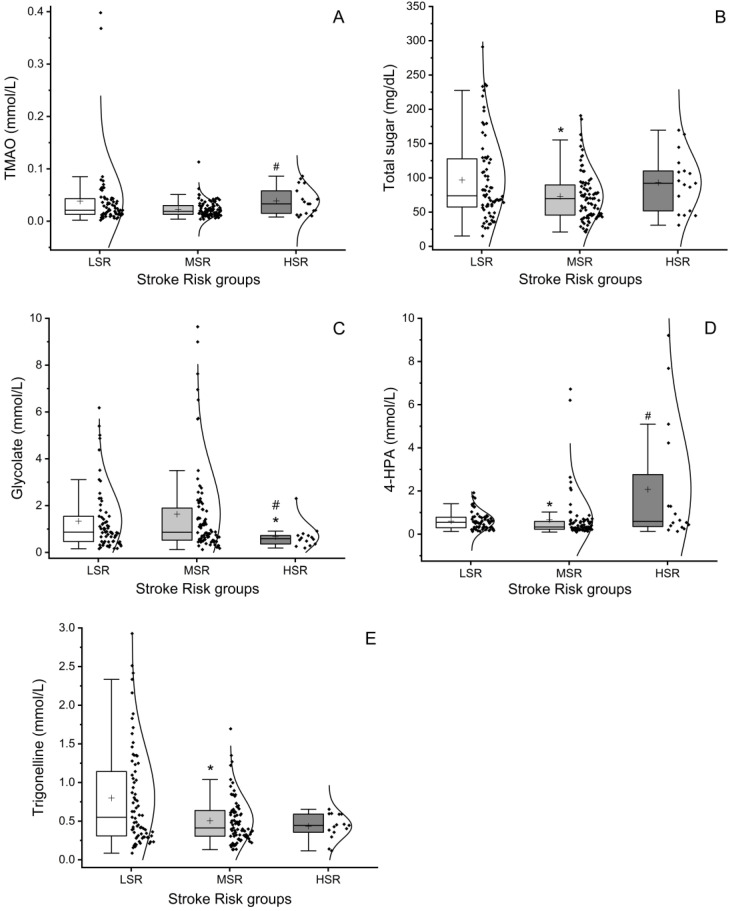
Box–violin–scatter plots representing the variations in absolute concentrations of trimethylamine *N*-oxide (TMAO). (**A**) total sugar, (**B**) glycolate, (**C**) 4-hydroxyphenylacetate (4-HPA), and (**D**) trigonelline (**E**) in the study groups, namely, low stroke risk (LSR, control), moderate stroke risk (MSR), and high stroke risk (HSR). ‘+’ in plots indicate average values. Statistical significance was analysed using the Kruskal–Wallis non-parametric test. Differences with the control group (LSR) are represented by ‘*’ (*p* ≤ 0.05) and group-pair, MSR and HSR, differences are represented by ‘#’ (*p* ≤ 0.05).

**Figure 3 ijms-25-07436-f003:**
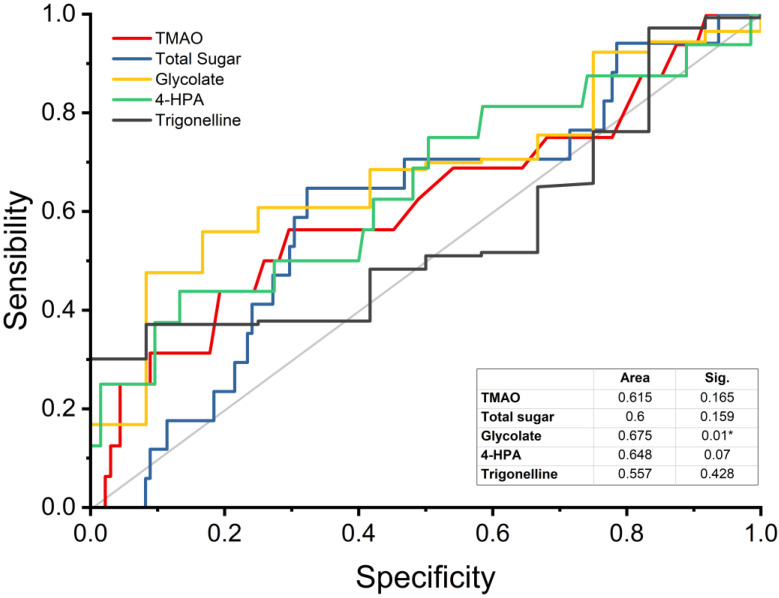
AUROC analysis for discriminating stroke risk using trimethylamine *N*-oxide (TMAO), total sugar, glycolate, 4-hydroxyphenylacetate (4-HPA), and trigonelline (* *p* ≤ 0.05).

**Table 4 ijms-25-07436-t004:** Correlations between changes in metabolite levels and age, and total number of cardiovascular diseases in stroke risk groups. 4-Hydroxyphenylacetate; 4-HPA. ▲/▼ = increase/decrease in metabolite levels; ns = nonsignificant correlation; r = Pearson’s chi-squared correlation coefficient. The statistical significance of correlations was analysed using Pearson’s chi-squared correlation coefficient (* *p* < 0.05; *** *p* < 0.001).

Data	Group	Total Sugar	r	Glycolate	r	4-HPA	r	Trigonelline	r
Comparison (metabolites) with age	LSR	ns	−0.033	ns	−0.143	ns	0.109	ns	−0.134
	MSR	ns	0.047	ns	−0.090	ns	0.110	ns	−0.111
	HSR	ns	−0.228	ns	−0.166	ns	0.196	ns	0.436
	All	ns	−0.118	ns	−0.082	▲ *	0.162	▼ ***	−0.258
Comparison (metabolites) with total number of cardiovascular diseases	LSR	ns	0.036	ns	−0.185	ns	−0.033	ns	0.005
	MSR	ns	0.261	ns	0.046	ns	0.077	ns	−0.054
	HSR	ns	−0.084	ns	−0.290	ns	0.181	ns	0.046
	All	ns	0.069	ns	−0.046	▲ *	0.178	ns	−0.095

**Table 5 ijms-25-07436-t005:** Correlations and comparisons between changes in metabolite levels and comorbidities or treatments in stroke risk groups. Comorbidities analysed are acute myocardial infarction, atrial fibrillation, and other arrhythmias. Pharmacological drug types studied were loop diuretics, proton pump inhibitors, medicines used to treat gout, and non-steroidal anti-inflammatory drugs (NSAIDs). 4-Hydroxyphenylacetate; 4-HPA. ▲/▼ = increase/decrease in metabolite levels when the indicated disease or treatment is present; ns = nonsignificant correlation or difference; r = Spearman’s rank correlation coefficient. Statistical significance of correlations was analysed using Spearman’s rank correlation coefficient (* *p* < 0.05). Statistical significance of metabolite values was analysed using Student’s *t*-test (* *p* < 0.05; *** *p* < 0.001).

Diseases/Treatments	Group	Total Sugar	r	Glycolate	r	4-HPA	r	Trigonelline	r
Comparison (metabolites) with and without Atrial fibrillation	LSR	ns	0.168	ns	0.084	ns	0.000	ns	0.077
	MSR	ns	0.140	ns	−0.096	ns	0.163	ns	0.011
	HSR	ns	−0.096	ns	−0.322	ns	0.041	ns	0.155
	All	ns	0.112	ns	−0.097	ns	0.102	▼ ***	−0.007
Comparison (metabolites) with and without loop diuretics treatment	LSR	ns	0.108	ns	0.128	ns	0.090	ns	0.163
	MSR	▲ *	0.225	ns	0.193	▲ *	0.251	ns	0.114
	HSR	ns	0.476	ns	0.401	ns	0.447	ns	0.452
	All	▲ *	0.186	ns	0.111	▲ *	0.221	ns	0.096
Comparison (metabolites) with and without proton pump inhibitor treatment	LSR	ns	0.229	ns	0.168	ns	0.264	ns	−0.060
	MSR	ns	−0.089	ns	0.110	ns	0.046	ns	−0.174
	HSR	ns	0.238	ns	0.330	ns	−0.136	ns	0.290
	All	▲ *	0.081	▲ *	0.105	ns	0.122	ns	−0.110
Comparison (metabolites) with and without Medicines used to treat gout	LSR	ns	.0.019	ns	−0.177	ns	0.003	ns	0.141
	MSR	ns	0.196	ns	−0.117	ns	−0.078	ns	−0.179
	HSR	ns	0.339	ns	0.494	ns	−0.142	ns	0.332
	All	ns	0.135	▼ *	−0.136	ns	−0.013	ns	−0.017
Comparison (metabolites) with and without NSAID treatment	LSR	ns	0.100	ns	0.002	ns	0.181	ns	0.003
	MSR	ns	0.006	ns	0.029	ns	0.177	ns	−0.275
	HSR	ns	0.399	ns	0.245	ns	0.310	ns	0.085
	All	ns	0.075	ns	0.019	▲ *	0.179	ns	−0.142

## Data Availability

The data that support the findings of this study are available on request from the corresponding author. The data are not publicly available due to restrictions of data protection law (Portugal) that concern because they contain information that could compromise the privacy of research participants.

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
