# Peer review of "Identification of Urine Metabolic Markers of Stroke Risk Using Untargeted Nuclear Magnetic Resonance Analysis"

_ijms, 2024, doi:10.3390/ijms25137436_

Round 1

Reviewer 1 Report

Comments and Suggestions for Authors

The research work addresses a quite interesting issue in terms of the attempt to identify specific biomarkers that could better describe and characterize the stroke risk in elderly patients. Overall, the manuscript is well structured, with a careful English spelling. The introduction is adjusted to the manuscript content, with relevant references being mentioned throughout the document. In this manuscript, the experimental design is well adjusted to the research purpose allowing to assess the different stroke risk groups.

However, there are some issues that we would further explanation from the authors, mainly in terms of the quantification procedure used in this manuscript and some considerations related with the statistical significance of the obtained results. Nevertheless, this can be easily overcome by the authors with adequate responses to some questions. In this way there are some points in the manuscript that I would appreciate a deeper explanation and/or revision from the authors:

1. Line 125, page 3: NMR quantification

The authors have used TSP and internal standard for quantification. This use as to be well explained and supported by literature since the TSP as been extensively mention in literature as interacting with macromolecules present in complex matrices affecting TSP peak intensity (in biofluids the impact of the proteins present in the sample is well documented to affect, although in urine the protein issue is minimized). Nevertheless, the major issue that I have in the “quantification” used in the manuscript is the fact that in NMR acquisition it was not used a relaxation delay with the duration of 5 x T1.

I would like the authors to defend their quantification procedure.

2. Line 210, page 7: trimethylamine NMR assignment

The authors have assigned the peak (3.26, singlet) as trimethyl amine. However, the databases that I use and my experience with complex matrices tells me that that peak is from TMAO.

How did the authors made the trimethylamine assignment? By metabolite spiking?

Please confirm carefully if the assignment is correct.

3. Line 240, page 8: Figure 2 and results discussion

This figure 2 and respective discussion requires improvement. Looking at the Figure and the box violin-scatter plots, I have difficulties in observing the statistical significance mentioned in the results discussion. For example, in line 249, the authors mention that Trimethylamine “…reduction in its level between MSR and HSR groups”. However, looking at the boxplots, the median value for HSR is higher than the one in MSR.
I have the same issues also for sugars content and 4-HPA (in the comparison between LSR and MSR). Can the authors describe better the approach used for the p-value calculation?

4. Line 275, page 10
: Comorbidities and Drugs Effect on Urine Metabolome

The authors in this section discuss the influence of certain diseases and pharmacological treatments in metabolites levels. This is well though and the experimental design applied allow that discussion. However, the authors mention strong correlations with r values lower than 0.2. In my opinion relevant correlations should only be considered above 0.5 (and even those should not be mention as strong). I suggest the authors to improve this results discussion by not extrapolating the obtained correlations (although not strong correlations, do the authors see meaningful relationships?).

In addition, did the authors tried to cluster the patients by age to see the impact of aged in the metabolism?

Author Response

Reviewer #1:

Does the introduction provide sufficient background and include all relevant references? Yes

Is the research design appropriate? Yes

Are the methods adequately described? Can be improved

Are the results clearly presented? Must be improved

Are the conclusions supported by the results? Must be improved

ANSWER: We rewrite some parts of the methods section to improve the methods description. The results and conclusion sections were rewritten according with the reviewers’ suggestions.

Comments and Suggestions for Authors

The research work addresses a quite interesting issue in terms of the attempt to identify specific biomarkers that could better describe and characterize the stroke risk in elderly patients. Overall, the manuscript is well structured, with a careful English spelling. The introduction is adjusted to the manuscript content, with relevant references being mentioned throughout the document. In this manuscript, the experimental design is well adjusted to the research purpose allowing to assess the different stroke risk groups.

However, there are some issues that we would further explanation from the authors, mainly in terms of the quantification procedure used in this manuscript and some considerations related with the statistical significance of the obtained results. Nevertheless, this can be easily overcome by the authors with adequate responses to some questions. In this way there are some points in the manuscript that I would appreciate a deeper explanation and/or revision from the authors:

  1. Line 125, page 3: NMR quantification

The authors have used TSP and internal standard for quantification. This use as to be well explained and supported by literature since the TSP as been extensively mention in literature as interacting with macromolecules present in complex matrices affecting TSP peak intensity (in biofluids the impact of the proteins present in the sample is well documented to affect, although in urine the protein issue is minimized). Nevertheless, the major issue that I have in the “quantification” used in the manuscript is the fact that in NMR acquisition it was not used a relaxation delay with the duration of 5 x T1.

I would like the authors to defend their quantification procedure.

ANSWER: Our study was carried out following standardized operating procedures for the production, conservation, and manipulation of the collected samples, which were recommended to us by the scientific support services of the company selling the equipment and by collaborating researchers. We also meticulously utilize the endorsed NMR pulse sequences for the metabolomic studies performed on human urine, as other authors performing similar studies did (doi-10.1016/j.jpba.2022.114885, doi-10.1021/acs.jproteome.8b00877, doi-10.1038/s41598-017-09203-3). In this sense, regarding the pulse sequence, we used a “noesygppr1d” Bruker library pulse sequence, with a 4μs t1 delay, 4s relaxation delay, and 2.72s acquisition (see pulse graphic below). To make this clearer, we introduced this information in the section ‘NMR data acquisition and processing’.

Also based on the SOP provided, we used the TSP as a reference standard for urine samples, since the elderly people included in the study did not have any kidneys’ filtering function impairment, and therefore, the presence of proteins and lipids in the urine is insignificant, has stated in several previous studies (doi-10.1016/j.xpro.2023.102181; doi-10.1007/s11306-014-0746-7; doi-10.1021/acs.jproteome.5b00885). Therefore, the possible interaction of TSP with macromolecules is meaningless.

  1. Line 210, page 7: trimethylamine NMR assignment

The authors have assigned the peak (3.26, singlet) as trimethyl amine. However, the databases that I use and my experience with complex matrices tells me that that peak is from TMAO.

How did the authors made the trimethylamine assignment? By metabolite spiking?

Please confirm carefully if the assignment is correct.

ANSWER: According with reviewer suggestion, the assignment was corrected, by analyzing the 1D-NOESY and 2D J-Res, HMBC and HSQC pool spectra. The 3.26ppm singlet is in fact TMAO. All related corrections were performed.

  1. Line 240, page 8: Figure 2 and results discussion

This figure 2 and respective discussion requires improvement. Looking at the Figure and the box violin-scatter plots, I have difficulties in observing the statistical significance mentioned in the results discussion. For example, in line 249, the authors mention that Trimethylamine “…reduction in its level between MSR and HSR groups”. However, looking at the boxplots, the median value for HSR is higher than the one in MSR.

I have the same issues also for sugars content and 4-HPA (in the comparison between LSR and MSR). Can the authors describe better the approach used for the p-value calculation?

ANSWER: For p-value calculation we performed the non-parametric statistical test of Kruskal-Wallis with an all pairwise analysis, using IBM SPSS software (version 29). We confirmed all the median values obtained, as well as all the corresponding p-values, and the values still match the ones previously presented. However, we recognize that in the way the results are described, some errors in interpretation may arise. Therefore, we rewrote this parts of the results and discussion sections, and Figure 2 was also improved.

  1. Line 275, page 10: Comorbidities and Drugs Effect on Urine Metabolome

The authors in this section discuss the influence of certain diseases and pharmacological treatments in metabolites levels. This is well though and the experimental design applied allow that discussion. However, the authors mention strong correlations with r values lower than 0.2. In my opinion relevant correlations should only be considered above 0.5 (and even those should not be mention as strong). I suggest the authors to improve this results discussion by not extrapolating the obtained correlations (although not strong correlations, do the authors see meaningful relationships?).

In addition, did the authors tried to cluster the patients by age to see the impact of aged in the metabolism?

ANSWER: In this analysis we looked for two types of data: 1) comparison of the mean values of the metabolites concentration between individuals with and without the referred diseases, and between people taking or not taking some medicines, using Students-T test; 2) correlation analysis between the metabolites levels and age, number of cardiovascular diseases, comorbidities and pharmacological treatments, with previously identified variations across the different stroke risk groups. We do not mention the documented correlations has ‘strong’, but as ‘significant’, due to the associated P-value. In the results section, we recognize that correlation coefficients bellow 0.2 are in fact low (lines 229-234, 257-259). We acknowledge the reviewer point but based on the significant differences found in the Students-T test analysis, we believe that some of the comorbidities and medicines might be confounders. Nonetheless, we revised and improved the writing of this section

Reviewer 2 Report

Comments and Suggestions for Authors

The work analyzes various molecules in urine through NMR spectroscopy in order to stratify stroke risk. The work looks interesting and well done. I recommend greater adherence to the journal format. Please add a figure that summerize the aim of the paper in order to make it more easy to read.

Comments on the Quality of English Language

Good

Author Response

Reviewer #2:

Does the introduction provide sufficient background and include all relevant references? Can be improved

Is the research design appropriate? Can be improved

Are the methods adequately described? Can be improved

Are the results clearly presented? Can be improved

Are the conclusions supported by the results? Can be improved

ANSWER: To improve the introduction and methods sections we rewrite some parts of these sections. Also, the results and conclusion sections were rewritten according with the reviewers’ suggestion.

Comments and Suggestions for Authors

Comments on the Quality of English Language

The work analyzes various molecules in urine through NMR spectroscopy in order to stratify stroke risk. The work looks interesting and well done. I recommend greater adherence to the journal format. Please add a figure that summarize the aim of the paper in order to make it more easy to read.

ANSWER: According with reviewer suggestion, a graphical abstract was added.

Reviewer 3 Report

Comments and Suggestions for Authors

The article submitted for review addresses a very important topic - the search for markers of myocardial infarction. The 1H NMR technique in the study of the metabolome continues to gain popularity, so this study fits in with current trends.

The introduction should have been expanded - why urine metabolites and not blood metabolites? What parameters are currently being assessed, what tests? What are the potential advantages of the authors' new method?

Methods section very well described. 

In the results and discussion, some difficulties of interpretation are posed by the sugars group:

- were the same sugars present in each patient?

- Was the ratio of sugars variable or constant?

- could the sugars have been derived only from dietary sources - as the Authors indicate, or did they also consider the possibility of their origin from glycosidic derivatives of compounds excreted in the urine?

- with such variability - origin, ratio composition - should they be considered as potential markers associated with disease?

Did the authors not consider supplementing the study with chemometric analysis? With such a large amount of data, it would have been a great addition. 

The manuscript should be adapted to the journal template.

Author Response

Reviewer #3:

Does the introduction provide sufficient background and include all relevant references? Must be improved

Is the research design appropriate? Yes

Are the methods adequately described? Yes

Are the results clearly presented? Can be improved

Are the conclusions supported by the results? Can be improved

ANSWER: The results and conclusion sections were rewritten according with the reviewers’ suggestions.

Comments and Suggestions for Authors

The article submitted for review addresses a very important topic - the search for markers of myocardial infarction. The 1H NMR technique in the study of the metabolome continues to gain popularity, so this study fits in with current trends.

The introduction should have been expanded - why urine metabolites and not blood metabolites? What parameters are currently being assessed, what tests? What are the potential advantages of the authors' new method?

Methods section very well described.

ANSWER: As to the question of ‘why urine metabolites and not blood metabolites’ we would like to highlight the fact that our results for plasma metabolites were already published (doi-10.3390/ijms242216173), and so with this study we intend to add new data to those previously reported. Since we had already reported the differences in circulating metabolites, we intended to verify whether there are also differences in urinary excreted metabolites.

Concerning the parameters currently being accessed to try to prevent stroke onset, they are based on controlling the comorbidities considered risk factors (lines 40-46). As to the already reported possible urine biomarkers for stroke they were also mentioned in the introduction section (lines 59-68). Nonetheless, we completed this information with recent non-metabolomic data obtained from non-metabolomic techniques.

Regarding the advantages of our new approach regarding stroke risk metabolomic studies, they are described in the first paragraph of the discussion section

In the results and discussion, some difficulties of interpretation are posed by the sugars group:

- were the same sugars present in each patient?

- Was the ratio of sugars variable or constant?

ANSWER: The measurement of total sugars by NMR involves the detection of chemically different protons existing in different sugars, which causes complex spectral patterns and signal overlap, which complicates the reliable identification and quantification of these molecules through this method (doi-10.1002/mrc.4525). Based on this, we can only assure the quantification of total sugar with precision. We believe that the sugars present in each elder were the same, because we were able to detect the same proton signals in all samples. However, due to the signal overlap, we could not determine with precision the ratio of each sugar, and so we gather all sugar signals and quantified the total sugar.

- could the sugars have been derived only from dietary sources - as the Authors indicate, or did they also consider the possibility of their origin from glycosidic derivatives of compounds excreted in the urine?

- with such variability - origin, ratio composition - should they be considered as potential markers associated with disease?

ANSWER: The urine NMR signals of total sugars does not allow us to discriminate between different carbohydrate molecules, thus it is not possible to figure out what these sugars are. In this sense and based on the uncertainty described in our results and discussion, as we referred in lines 439-442, we believe that we cannot consider total sugars a stroke biomarker. A deeper analysis using different protocols or techniques must be done. On behalf of this, a specific study of these compounds should be drawn, to highlight its source and the influence of diet and water intake.     

Did the authors not consider supplementing the study with chemometric analysis? With such a large amount of data, it would have been a great addition.

ANSWER: Concerning chemometric analysis (multivariate analysis), initially, we were more interested in univariate analysis, because this allows us to understand the distribution of values for the variables that this type of analysis considers most important, in this case stroke risk and metabolites, which allows us to analyze the relationship between them, sometimes carrying out paired analyzes or analyzing the crosstalk of a variable in a relationship between two other variables. We think this it will be interesting to perform multivariable analysis with a cohort with bigger amount on individuals, because ours is still limited to perform this kind of analysis..

In fact, when we begin the biomarkers analysis in related areas by performing multivariate analyses, such as multiple linear regression models or principal component analysis, and we feel that, in many cases, "the relationships" are unclear or that could be the result of mathematical constructs. On the other hand, if the multivariate analysis is not carried out very carefully, it may lead to underestimating or giving less value to variables that may be important, and the opposite can also occur. In this context, we first decided to move forward with univariate analysis to better understand the most important variables, which does not exclude us from carrying out multivariate analyzes in the future but with more data related to other types of molecules (lipids or proteins, for example).

The manuscript should be adapted to the journal template.

ANSWER: The manuscript was adapted to the journal template.

Round 2

Reviewer 1 Report

Comments and Suggestions for Authors

I recognize the effort and work of the authors to address the questions and suggestions made by the reviewers. Therefore, my recommendation is to accept the manuscript. 

Nevertheless, there are some issues in the manuscript related with replacing “Trimethylamine” by ”Trimethylamine N-oxide (TMAO)”. The authors must check carefully if they have replaced all “trimethylamine” by ”trimethylamine N-oxide” – I found at least in lines 214, 216 and 354 references of trimethylamine. Also, in all cases, the authors must use italic lettering as follows: “trimethylamine N-oxide”.

Author Response

COMMENTS:

I recognize the effort and work of the authors to address the questions and suggestions made by the reviewers. Therefore, my recommendation is to accept the manuscript. 

Nevertheless, there are some issues in the manuscript related with replacing “Trimethylamine” by ”Trimethylamine N-oxide (TMAO)”. The authors must check carefully if they have replaced all “trimethylamine” by ”trimethylamine N-oxide” – I found at least in lines 214, 216 and 354 references of trimethylamine. Also, in all cases, the authors must use italic lettering as follows: “trimethylamine N-oxide”.

RESPONSE:

Thank you for pointing this out this issue. Therefore, all the “trimethylamine” were carefully checked and replaced by “trimethylamine N-oxide”, using italic lettering as recommended. However, in lines 353, 355 and 356 we intent to talk about trimethylamine as a body precursor of TMAO, and the metabolic reactions that lead to synthesis of TMAO, so in this lines the “trimethylamine” remain as in the previous version of the manuscript.

Reviewer 3 Report

Comments and Suggestions for Authors

The Authors have answered all questions and improved the manuscript. Im my opiniom it can be now accepted.

Author Response

COMMENTS:

The Authors have answered all questions and improved the manuscript. In my opinion it can be now accepted.

RESPONSE:

The reviewer states that we already answered all the questions in the previous round.